# The Relationship between Peer Attachment and Aggressive Behavior among Chinese Adolescents: The Mediating Effect of Regulatory Emotional Self-Efficacy

**DOI:** 10.3390/ijerph18137123

**Published:** 2021-07-02

**Authors:** Haitao Liu, Kai Dou, Chengfu Yu, Yangang Nie, Xue Zheng

**Affiliations:** 1School of Psychology, South China Normal University, Guangzhou 510631, China; soeliuht0919@gzhu.edu.cn; 2Department of Psychology and Research Center of Adolescent Psychology and Behavior, School of Education, Guangzhou University, Guangzhou 510006, China; psydk@gzhu.edu.cn (K.D.); cnwhycf@163.com (C.Y.)

**Keywords:** adolescents, aggressive behavior, peer attachment, regulatory emotional self-efficacy

## Abstract

This study aimed to test the association between peer attachment and aggressive behavior, as well as the mediating effect of regulatory emotional self-efficacy on this relationship. A total of 1171 (582 male, 589 female) Chinese adolescents completed self-reported questionnaires that assessed peer attachment, regulatory emotional self-efficacy, and aggressive behavior. Path analysis showed that the negative association between peer attachment and adolescent aggressive behavior was mediated by self-efficacy in managing negative emotions. However, the mediating effect of self-efficacy in expressing positive emotions was nonsignificant. Moreover, there was no significant difference in the indirect paths mentioned above between male and female respondents. These findings highlight self-efficacy in managing negative emotions as a potential mechanism linking peer attachment to adolescent aggressive behavior.

## 1. Introduction

The frequent occurrence of campus violence in recent years has not only caused widespread concern among the media and public but also prompted researchers to focus on the internal mechanisms of juvenile violent aggression. Aggressive behavior refers to an individual’s act of inflicting harm on others in various forms [1] and is often associated with low levels of self-control [2], insecure parental attachment [3], and family dysfunction [4,5]. It has been reported that 17.9% of Chinese adolescents have displayed physically aggressive behaviors toward their peers once or more during the last 12 months [6]. Similarly, in a recent study population including 15,975 Chinese adolescents, it was shown that about 26% of them display aggression [7].

However, as self-consciousness gradually increases in adolescence, teenagers’ dependence on their parents begins to decrease [8,9]. Expanding on previous research, this study’s primary purpose is to use the perspective of self-evaluation to examine whether peer attachment can predict aggressive behavior through regulatory emotional self-efficacy. As teenagers have a strong need for affinity, when their parental dependence decreases, they actively seek to establish more emotional connections with their peers, which are expressed through intimacy, warmth, and support [10]. Therefore, peer attachment might be an important predictor of aggressive behavior among adolescents.

### 1.1. Peer Attachment and Aggressive Behavior

Adolescence is the transitional period between childhood and adulthood. Individuals at this stage of development are prone to impulsivity and poor self-control; thus, they are more likely to have aggressive responses [11]. Studies have shown that adolescents’ aggressive behavior might be impacted by peer factors [12,13]; peers have both positive and/or negative effects on adolescents’ behavior [14,15].

These relationships can be understood in terms of peer attachment, which refers to an emotional connection that an individual forms in a long-term relationship with a peer. Adolescents not only gain intimacy, warmth, and social support from this relationship [10], but also use it to compensate for the negative influence of their families [16]. Studies have observed that poor attachment relationships with peers are an important cause of adolescents’ problem behaviors [17] and that improving peer attachment can improve adolescents’ prosocial behavior [18]. Another study found that peer exclusion can cause adolescents to lose their sense of control over their environment, leading to aggressive behavior [19]. Individuals with insecure attachments are more likely to be emotionally distant from others, since these individuals may have stronger emotional responses to conflicts in intimate relationships, which can lead to deviant behaviors and hostility. Therefore, it can be inferred that a lack of peer attachment might be a major cause of adolescent aggression [20]. 

### 1.2. Regulatory Emotional Self-Efficacy as a Mediator

Forming a secure attachment relationship with peers is an important factor in evaluating the self positively [21]. According to the social network theory, individuals establish emotional ties with many important people over the course of their lives, and these people play a role in providing important psychological needs, such as emotional support and affirmation [22]. Previous studies have verified that attachment relationships indirectly affect adolescents’ aggressive behavior through aspects of self-evaluation (e.g., self-control, cognitive distortion, and self-esteem) [2,5,23]. Among these factors, regulatory emotional self-efficacy (a specific type of self-evaluation) has been proven to have an important influence on adolescents’ problem behaviors [24]. 

Regulatory emotional self-efficacy refers to an individual’ s degree of self-confidence that they can successfully manage their emotional state, including the two dimensions of self-efficacy: managing negative emotions (NEG) and expressing positive emotions (POS). The former refers to faith that one can improve one’s mood in a timely manner when one encounters a frustrating event, and the latter refers to faith that one can express happiness, wellbeing, and pride when experiencing success or joy [24,25]. On the one hand, the more confident adolescents feel about coping with their negative emotional state, the higher the levels of self-control to define the impact of the negative emotion, which decreases the rate of aggressive behavior [26]. On the other hand, adolescents with high self-evaluation are more likely to adopt positive coping strategies that result in a decrease in negative emotions and an increase in subjective wellbeing, and thus, they are less prone to aggressive behavior [27]. Inversely, adolescents without confidence in emotion management tend to adopt more of an “avoidance” coping mode, leading to more serious passive emotions and more aggressive behaviors [28]. Based on this, it is believed that adolescents with better regulatory emotional self-efficacy can more effectively respond to interpersonal conflicts, regulate negative emotions, and abstain from aggressive behavior.

Establishing safe emotional connections with peers also enhances adolescents’ ability to regulate emotions and curb aggressive behavior. Peers can meet adolescents’ emotional needs during the adolescent developmental period, enabling them to receive peer support and help when they suffer setbacks or pressure, as well as providing them with an avenue to share positive emotions [29]. Thus, the ability to establish stable and safe peer attachment relationships is conducive to the formation of positive regulatory emotional self-efficacy among adolescents. Numerous studies have confirmed that regulatory emotional self-efficacy can alleviate impulsive problem behaviors caused by negative emotions that result from encountering bad situations (e.g., stress and depression) [30,31]. Therefore, it has been speculated that when adolescents cope with emotional problems, the two aspects of regulatory emotional self-efficacy (i.e., POS and NEG) may play a mediating role in peer attachment and influence their aggressive behavior. Hence, the current study proposes the following hypothesis. 

Hypothesis: Regulatory emotional self-efficacy (including POS and NEG) mediates the association between peer attachment and adolescent aggressive behavior. The hypothesis is visually presented in Figure 1.

The link between peer attachment and adolescent aggressive behavior via regulatory emotional self-efficacy might be moderated by gender. For example, Li and Guo [32] found that the influence of roommates on aggressive behavior was more pronounced among male than female adolescents. Therefore, it is necessary to further test whether the direct and indirect paths of peer attachment to adolescent aggressive behavior are moderated by gender.

To the best of our knowledge, no prior study on peer attachment simultaneously explores the effect of the two dimensions of regulatory emotional self-efficacy (POS and NEG) on aggressive behavior among adolescents. In addition, existing research on mental health and peer influence is focused on negative peer behavior, whereas the current study uses the concept of peer attachment to focus on the emotions of adolescents and to explore adolescent aggressive behavior. Based on existing theories and research, this study aims to investigate: (1) whether peer attachment is negatively associated with aggressive behavior, and (2) “how” (i.e., mediating mechanisms) peer attachment affects aggressive behavior and its gender difference. It extends previous research and provides meaningful implications to reduce aggressive behavior among adolescents.

## 2. Materials and Methods

### 2.1. Participants and Procedures

We recruited participants from three secondary schools in Guangzhou city, southern China. The sample was first stratified by type of school: junior high schools, senior high schools, and complete middle schools (which combine junior and senior high school). Random cluster sampling was used to select two classes from each grade in each school. The authenticity, independence, and integral nature of all answers as well as confidentiality of the information collected were emphasized by well-trained graduate students of psychology. A total of 1200 adolescents participated in this study with the informed consent of school principals, parents, and the adolescents themselves. Twenty-nine questionnaires were found invalid for the following reasons: (a) more than 10% missing data; and (b) inconsistent responses for homogeneous items or consistent responses for reverse items. In total, 1171 valid questionnaires were included in the analyses. Of these, 547 respondents were junior middle school students (grade 7, 8, and 9 participants numbering 182, 218, and 147, respectively) and 624 were senior middle school students (grade 10, 11, and 12 participants numbering 205, 213, and 206, respectively). Regarding gender distribution, 49.70% of participants were male (*n* = 582) and 50.30% were female (*n* = 589). The participants ranged in age from 11 to 19 years, with an average age of 14.86 years (*SD* = 1.87 years). Table 1 shows the characteristics of the sample.

The study was conducted according to the guidelines of the Declaration of Helsinki, and approved by the Ethics in Human Research Committee, Department of Psychology, Guangzhou University (protocol code: GZHU 2019008, date of approval: 27 May 2019).

### 2.2. Measures

#### 2.2.1. Peer Attachment

The revised version of the Inventory of Parent and Peer Attachment was used to measure participants’ attachment to peers. This scale consists of 25 items rated on a 5-point Likert scale (from “1 = never” to “5 = always”) assessing the magnitude of trust, communication, and alienation toward peers. A sample item is “my peer tries to understand me when I’m angry about something.” Better peer attachment is indicated by higher scores on trust and communication and lower scores on alienation. This scale has been previously shown to have adequate psychometric properties when applied to Chinese people [21]. The Cronbach’s alpha for this scale is 0.92.

#### 2.2.2. Regulatory Emotional Self-Efficacy

The Chinese version of the regulatory emotional self-efficacy scale was used to assess participants’ evaluations of their own competency in regulating their emotions [24,33]. This scale consists of 17 items rated on a 5-point Likert scale (from “1 = not like me at all” to “5 = very much like me”), measuring two dimensions: POS and NEG. A sample item is “I express my excitement when I come across people or things I like.” A high score indicates that participants believe they are able to express positive emotions and manage negative emotions well. This scale has been translated into Chinese and has shown adequate psychometric properties when applied to Chinese adolescents [24]. The Cronbach’s alphas for the POS and NEG subscales are 0.86 and 0.89, respectively. 

#### 2.2.3. Aggressive Behavior

The aggression tendency dimension of the Behavioral Tendency Questionnaire for Adolescent Behavior Problems compiled by Zhang et al. [34] was used to measure adolescents’ aggressive behavior during the past six months. A sample item is “I often hit the wall or something like that when I’m bored.” This question consists of six items measured on a 5-point Likert scale (from “1 = never” to “5 = always”), where a high score indicates a strong aggression tendency. The Cronbach’s alpha for the aggression tendency dimension is 0.83.

### 2.3. Data Analyses

SPSS 21.0 (IBM, Chicago, IL, USA) and Mplus Version 7.4 (Muthén & Muthén, Los Angeles, CA, USA) were used to analyze the data. First, we examined group differences of the study variables between junior school versus high school students using an independent-sample *t* test. Second, we performed a correlation analysis to determine the correlations between the variables. Third, we tested mediation effects using structural equation modeling with maximum likelihood estimation and bootstrapping with 1000 replications in Mplus 7.4 [35]. According to Hoyle’s suggestion [36], the model fit is considered acceptable when *χ*^2^/*df* < 5, CFI > 0.90, TLI > 0.90, RMSEA < 0.08, and SRMR < 0.08. 

## 3. Results

### 3.1. Descriptive Statistics

Table 2 shows group differences of the study variables between junior school versus high school students. The results showed that junior school students’ parents had significantly higher levels of education than parents of high school students. However, there were no significant differences in peer attachment, POS, NEG, and aggressive behavior between junior school versus high school students.

Table 3 shows the means, standard deviations, and correlations of the study variables. Age was negatively correlated with peer attachment and positively correlated with aggressive behavior. Meanwhile, gender (0 = female, 1 = male) was negatively correlated with peer attachment and POS and positively correlated with NEG and aggressive behavior. In addition, peer attachment was found to be positively correlated with POS and NEG. Peer attachment was negatively correlated with aggressive behavior. Both POS and NEG were negatively correlated with aggressive behavior.

### 3.2. Mediation Analyses

Given that age, gender, father’s education level, and mother’s education level were associated with adolescent regulatory emotional self-efficacy and/or aggressive behavior, these variables were included as covariates in our statistical analyses. The initial model was a saturated model. We removed nonsignificant paths: (1) the paths of father’s education level and mother’s education level to POS, NEG, and aggressive behavior; (2) the path of gender to NEG; and (3) the path of POS to aggressive behavior. Because the path of POS to aggressive behavior was nonsignificant, we removed the variable for POS from the model. The final model represented in Figure 2 revealed an excellent fit to the data: *χ*^2^ = 0.26, *df* = 1, *χ*^2^/*df* = 0.26, CFI = 1.00, RMSEA = 0.000 (90% CI (0.000, 0.062)), and SRMR = 0.003.

The results showed that: (1) Peer attachment was significantly associated with NEG (*b* = 0.40, SE = 0.04, *β* = 0.33, *t* = 10.44, *p* < 0.001, 95% CI [0.33, 0.47]); (2) NEG was significantly associated with aggressive behavior (*b* = −0.21, SE = 0.04, *β* = −0.19, *t* = −5.61, *p* < 0.001, 95% CI [−0.28, −0.14]); and (3) the residual effect of peer attachment on aggressive behavior was significant (*b* = −0.11, SE = 0.04, *β* = −0.08, *t* = −2.55, *p* < 0.05, 95% CI [−0.20, −0.02]). (4) Moreover, age was significantly associated with NEG (*b* = 0.20, SE = 0.04, *β* = 0.14, *t* = 5.02, *p* < 0.001, 95% CI [0.12, 0.27]) and aggressive behavior (*b* = 0.26, SE = 0.05, *β* = 0.16, *t* = 5.87, *p* < 0.001, 95% CI [0.18, 0.35]); and (2) gender was significantly associated with aggressive behavior (*b* = 0.03, SE = 0.01, *β* = 0.07, *t* = 2.76, *p* < 0.01, 95% CI [0.01, 0.06]); (5) Finally, the bootstrapping analyses indicated that NEG significantly mediated the relationship between peer attachment and aggressive behavior (indirect effect = −0.08, SE = 0.02, *β* = −0.06, *t* = −4.70, *p* < 0.001, 95% CI [−0.12, −0.05]).

Finally, we tested whether the mediating model examined above was moderated by gender. Specifically, we tested whether the indirect path “peer attachment → NEG → adolescent aggressive behavior” is moderated by gender and whether the residual effect of peer attachment on aggressive behavior was moderated by gender. The results indicated that only the residual effect of peer attachment on aggressive behavior was moderated by gender. We conducted a simple slopes test. As depicted in Figure 3, peer attachment was significantly associated with aggressive behavior among male adolescents (*b* = −0.28, SE = 0.06, *t* = −4.86, *p* < 0.001, 95% CI [−0.40, −0.17]). However, this link was not significant among female adolescents (*b* = 0.05, SE = 0.06, *t* = 0.92, *p* = 0.359, 95% CI [−0.06, 0.16]).

## 4. Discussion

Previous research on adolescent peer relationships has shown that establishing and maintaining a good attachment relationship with peers is beneficial to adolescents’ healthy development, while poor peer attachment often fosters harmful behavior in adolescents, such as aggressive behavior [37,38]. This study explores the influence mechanism of adolescents’ peer attachment on aggressive behavior, demonstrating that safe peer attachment can prevent aggressive behavior in adolescents and that this effect can be indirectly managed through the mediator of NEG. 

In the present study, adolescents with strong peer attachment show less aggressive behavior. This result confirms our hypothesis, and is consistent with the work of Yuksek and Solakoglu [17], which demonstrates that peers are important for adolescents and that peer attachment has significant predictive effects on adolescents’ aggressive behavior. Krause et al. [39] found that it is more difficult for adolescents who believe that they are neglected by their peers to get along with classmates, and that they usually experience negative emotions, which leads to problem behaviors. Some researchers have suggested that adolescents’ empathy has a negative relationship with aggressive behavior, since adolescents with lower empathy often have difficulties in understanding their peers’ emotions, which consequently tends to exacerbate aggressive behavior [37]. 

The results of this study also suggest that a high level of peer attachment is conducive to the development of good social behavior, while a low level of peer attachment manifesting as a lack of necessary trust and communication with peers is positively associated with aggressive behavior. That is, when adolescents are attached to their peers, they actively maintain interpersonal relationships and confide more in their peers; in turn, their peers provide them advice and support. Adolescents with weak peer attachment may be unable to confide in their peers when confronted with pressure due to life or academic pursuits, and thus tend to vent their negative emotions in an unhealthy way, such as aggressive behaviors. However, as Bronfenbrenner’s ecological systems theory explains [40], if adolescents cannot receive support and warmth from their peer systems, these systems can be replaced by others. This illustrates the importance of parents and teachers paying attention to the emotional factors in adolescents’ peer relationships and intervening promptly if they notice a lack of peer support and warmth. 

More importantly, this study indicates that the NEG dimension of regulatory emotional self-efficacy plays a mediating role between peer attachment and adolescent aggressive behavior, which partially confirms our hypothesis. As shown by previous research [41], regulatory emotional self-efficacy is conducive to reducing adolescents’ aggressive behavior. Adolescents who believe they can deal with emotional problems can effectively control the interference of negative emotions and behave appropriately when confronted with external stressors. Papadakaki et al. [42] pointed out that individuals who are not confident in their abilities are prone to violence. According to the internal working model of attachment, attachment can affect individuals’ cognition of their own abilities, especially in early childhood, when the parenting style the child is exposed to forms a fixed cognitive schema. Therefore, it is believed that when peers show the same behaviors and attitudes as early childhood caregivers, the same cognitive schema of adolescents are activated, thereby affecting their self-efficacy. The findings of this study show that NEG has a mediating effect on aggressive behavior. This implies that individuals with low peer attachment may be influenced by their peers into believing that they lack the ability to regulate negative emotions and, thus, choose extreme behaviors, such as indulgence or suppression of their own negative emotions, finally generating aggressive behavior against others or themselves. 

This study also found that the mediating effect of POS between peer attachment and adolescents’ aggressive behavior was nonsignificant. This finding suggests that the effect of hiding positive emotions on adolescents’ aggressive behavior is not as powerful as suppressing negative emotions. It should be noted that this study examined only the mediating effect of regulatory emotional self-efficacy between peer attachment and adolescents’ aggressive behavior. Future research should further explore whether regulatory emotional self-efficacy mediates the effect of peer attachment on other behavioral and emotional symptoms.

This study also found no significant difference in the indirect paths mentioned above between male and female participants. Furthermore, the residual effect of peer attachment on aggressive behavior was moderated by gender. This finding is consistent with previous research [36] suggesting that the magnitude and direction of peer influence depend on gender. Future research should explore other moderating variables in the indirect path of peer attachment to aggressive behavior via regulatory emotional self-efficacy.

## 5. Implications

The present study makes the following contributions to the field. First, it constructs and confirms a mediating model with regulatory emotional self-efficacy as an intermediary, which reveals the development mechanism of adolescents’ aggressive behavior. Second, in contrast to most existing research on mental health and peer influence, which starts from the perspective of poor peer behavior learning, this study focuses on the emotions of adolescents and explores aggressive behavior using the concept of peer attachment. Third, it demonstrates that regulatory emotional self-efficacy plays an indispensable role in adolescents’ development. This is important, since past research has emphasized emotional regulation but neglected adolescents’ self-evaluation of their self-regulation ability. This finding strongly suggests that teachers, parents, social service workers and so on should actively cultivate adolescents’ regulatory emotional self-efficacy. 

For educational practice, courses or activities on emotion education should be used as a vehicle to convey emotional regulation knowledge and skills to adolescents, guide them in regulating their own emotions, and gradually enhance their regulatory emotional self-efficacy. When adolescents, under the guidance of teachers and parents, are able to face and accept their emotions, try to describe their feelings in words, learn to broaden their horizons, and reinterpret the meaning of certain events, it enables them to take control of their emotions. They then become more confident in their ability to regulate their emotions. Increase in adolescents’ regulatory emotional self-efficacy helps improve their emotional state; more importantly, they gain a sense of control over their daily emotional experiences and their self-confidence increases. This good feeling, in turn, extends to other aspects of self-understanding and decreases externalizing behaviors.

## 6. Limitations

Despite its contributions, this study has several limitations. First, although the reliability and validity analyses showed that the scales met the requirements of psychological statistics, it is difficult to avoid the reliability problem caused by self-reporting. Future research should aim to collect data from multiple informants and methods (e.g., teacher assessment and peer nomination). Second, the cross-sectional design cannot fully reveal the causal relationship between peer attachment and externalization of problem behaviors in adolescents. Therefore, future research should use vertical tracking technology to explore the relationship between peer attachment and aggressive behavior. Third, the present study focused on the unidirectional relationship between peer attachment and aggressive behaviors. However, Charalampous et al. [43] found that peer attachment also influences bullying/victimization among adolescents. Therefore, it would be meaningful to examine the bidirectional effect of peer attachment and aggressive behaviors in future research. Finally, this study considered only one mediator (i.e., regulatory emotional self-efficacy). Future research should explore other mediators or moderators in the association between peer attachment and aggressive behaviors.

## 7. Conclusions

The current study demonstrated that peer attachment is a crucial factor in adolescent aggressive behavior via regulatory emotional self-efficacy. Such information would be of great value to those in the helping professions working with adolescents.

## Figures and Tables

**Figure 1 ijerph-18-07123-f001:**
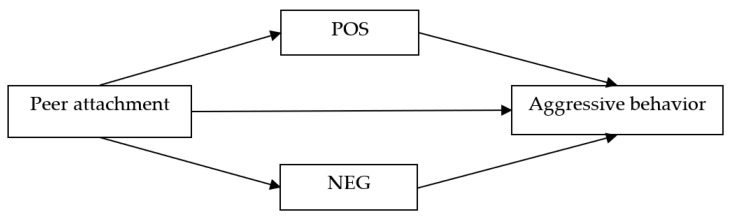
The proposed conceptual model of peer attachment, regulatory emotional self-efficacy, and aggressive behavior. POS, self-efficacy in expressing positive emotions; NEG, self-efficacy in managing negative emotions.

**Figure 2 ijerph-18-07123-f002:**
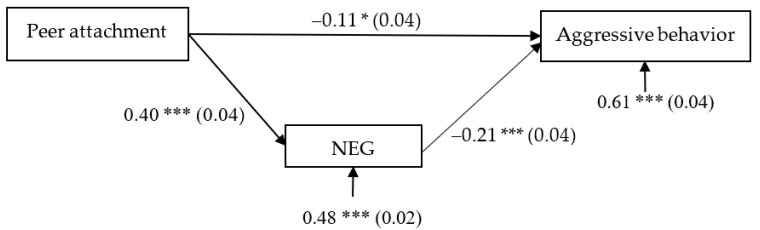
Model of the mediating role of regulatory emotional self-efficacy between peer attachment and aggressive behavior. NEG, managing negative emotions. Values outside parentheses are unstandardized coefficients, and those within are standard errors. * *p* < 0.05; *** *p* < 0.001.

**Figure 3 ijerph-18-07123-f003:**
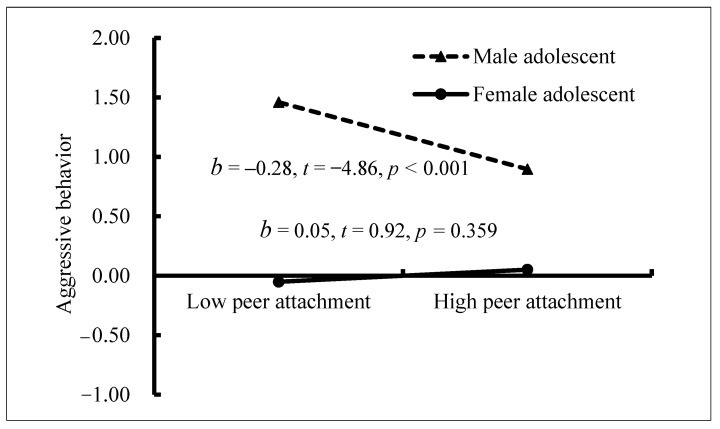
Aggressive behavior among adolescents as a function of peer attachment and gender.

**Table 1 ijerph-18-07123-t001:** Demographic information about the participants.

Variables	Female	Male	*n (%)*
Grade 7	97	85	182 (15.54)
Grade 8	100	118	218 (18.62)
Grade 9	71	76	147 (12.55)
Grade 10	119	86	205 (17.51)
Grade 11	101	112	213 (18.19)
Grade 12	101	105	206 (17.59)

**Table 2 ijerph-18-07123-t002:** Group differences of study variables between junior school versus high school students.

Variables	Junior School Students		High School Students		*t*	Cohen’s *d*
	Mean	SD	Mean	SD		
Age	13.18	1.00	16.34	1.02	—	—
Father’s education level	2.62	0.74	2.37	0.72	5.84 ***	0.34
Mother’s education level	2.46	0.78	2.15	0.71	7.08 ***	0.42
Peer attachment	2.78	0.64	2.78	0.57	0.08	0.00
POS	4.01	0.83	4.09	0.73	−1.67	0.10
NEG	3.42	0.76	3.39	0.71	0.77	0.04
Aggressive behavior	1.54	0.81	1.62	0.83	−1.55	0.10

Note: *n* = 1171. POS, self-efficacy in expressing positive emotions; NEG, self-efficacy in managing negative emotions; SD, standard deviation. *** *p* < 0.001.

**Table 3 ijerph-18-07123-t003:** Means, standard deviations, correlations, and reliabilities among the variables.

Variables	Age	Gender	FEL	MEL	PA	POS	NEG	AB
Age	1.00							
Gender	0.04	1.00						
FEL	−0.23 ***	−0.01	1.00					
MEL	−0.25 ***	−0.03	0.66 ***	1.00				
PA	−0.07 *	−0.16 ***	0.11 ***	0.12 ***	1.00			
POS	0.01	−0.22 ***	0.08 **	0.06 *	0.38 ***	1.00		
NEG	−0.03	0.08 **	0.09 **	0.09 **	0.31 ***	0.40 ***	1.00	
AB	0.09 **	0.16 ***	−0.03	−0.01	−0.17 ***	−0.16 ***	−0.20 ***	1.00
**Mean**	14.86	—	2.49	2.3	2.78	4.05	3.4	1.58
**SD**	1.87	—	0.74	0.76	0.6	0.78	0.73	0.82

Note: *n* = 1171. Gender was dummy coded as 0 (=female) and 1 (=male). FEL, father’s education level; MEL, mother’s education level; PA, peer attachment; AB, aggressive behavior. Father’s and mother’s education level were rated on a 4-point scale (“1 = primary school or below” to “4 = graduate degree”). POS, self-efficacy in expressing positive emotions; NEG, self-efficacy in managing negative emotions; SD, standard deviation. * *p* < 0.05; ** *p* < 0.01; *** *p* < 0.001.

## Data Availability

The data that support the findings of this study are available from the corresponding author, upon reasonable request.

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
