# Peer review of "The Relationship between Peer Attachment and Aggressive Behavior among Chinese Adolescents: The Mediating Effect of Regulatory Emotional Self-Efficacy"

_ijerph, 2021, doi:10.3390/ijerph18137123_

Round 1
Reviewer 1 Report
I would like to thank the Authors for all changes and improvements they have done. I have no further comments.
Author Response
- Thank very much for your constructive suggestions.
Reviewer 2 Report
The paper is now more clear and interesting for the research in this area
Author Response
- Thank very much for your constructive suggestions
Reviewer 3 Report
I appreciate the Authors addressing each of my concerns. I have no further comments.
Author Response
Thank very much for your constructive suggestions.
This manuscript is a resubmission of an earlier submission. The following is a list of the peer review reports and author responses from that submission.
Round 1
Reviewer 1 Report
The authors submitted a manuscript evaluating relationships between self-reported peer attachment, aggressive behavior, and regulatory emotional self-efficacy (RESE) within Chinese adolescents. The authors proposed two hypotheses: 1) peer attachment would negatively predict aggressive behavior, and 2) two components of RESE (managing negative emotions and expressing positive emotions) would mediate this negative relationship. To evaluate these relationships and test these hypotheses, the authors considered cross-sectional survey responses from 1108 Chinese students (549 junior high, 559 high school), calculated a series of correlation coefficients, and conducted a parallel mediation analysis. Results partially supported the authors’ hypotheses: as expected, peer attachment was negatively associated with aggressive behavior, and managing negative emotions emerged as a mediator of this relationship. However, expressing positive emotions did not mediate the relationship between peer attachment and aggressive behavior.
The authors are commended for exploring relationships between potentially impactful social behaviors during adolescence. Adolescence is considered by many to be a developmental phase in which relationships can powerfully influence emotional and behavioral outcomes, so continued multicultural research into relationships like those described in this manuscript is important and warranted. However, the manuscript as written possesses both major and minor limitations that negatively impact the paper’s suitability for publication in International Journal of Environmental Research and Public Health.
Major Limitations
First, the authors are encouraged to improve the reporting and justification behind the mediation analysis conducted in this manuscript. The following issues should be addressed:
- Conducting mediation analysis on cross-sectional data is not universally supported and could produce biased estimates of relationships and effects (see Maxwell & Cole, 2007, Psychological Methods). Because of this, drawing causal inferences from such an approach is discouraged. The authors do well to mention their cross-sectional methodology as a limitation towards the end of the Discussion and to encourage longitudinal follow-up studies. However, the manuscript also includes hints at causality in other places. One example of this occurs on page 6, where the authors state: “The results of this study also suggest that a high level of peer attachment is conducive to the development of good social behavior, while a low level of peer attachment can manifest as a lack of necessary trust and communication with peers, alienation, and being more prone to aggressive behaviors.” Another occurs on page 3, where the authors state “Based on the previous research, this study examines whether peer attachment can affect aggressive behaviors by first affecting RESE as a form of self-evaluation.” The authors are encouraged to critically review their manuscript and to be less forceful at inferring causal relationships from their design and findings.
- The authors are strongly encouraged to produce a series of two figures that lay out the following: 1) a conceptual model of the mediation analysis that includes no coefficients/statistics, but demonstrates the expected, hypothesized direct and indirect relationships; and 2) the full mediation model that visually depicts the indirect/a and b, direct/c, and total/c’ effects described in the Results, as well as the effects of the covariates (age and gender) on the outcome.
Second, the authors are encouraged to more thoroughly reinforce the importance, novelty, and contribution of this manuscript to the literature and to more comprehensively answer the “So what?” question that readers may ask when reviewing their work. Some ideas on how this can be done include the following:
- The authors say towards the end of the Introduction that, although the relationship between peer attachment and aggression has been already established, “merely examining this relationship is not sufficient to create a practical application.” Application of what?
- Is there something unique about Chinese adolescents that make them particularly vulnerable to developing aggressive behavior than adolescents from other developed countries?
- How is this manuscript advancing understanding of adolescent development within the aims and scope of this journal: environmental health sciences, public health, mental health, global health, children’s health, etc.?
Minor Limitations
Several questions regarding the Materials and Methods section require clarification:
- What made a questionnaire “valid” in this particular study? In other words, what led to the reduction in sample size from 1200 to 1108?
- The authors report distributing questionnaires to “middle school students from the first to the third grade” but then report collecting data from “junior high school and high school students”. Is this a typo on the authors’ part?
- What were the months/years of data collection?
- What was the “cluster” level factor for the sampling method employed by the authors?
- Since participants were not adults, was assent collected as a part of the protocol? And were participants compensated?
- How are the self-report measures scored, and what are the ranges of possible scores? Further, are there thresholds that indicate things like “high,” “low,” or “clinically significant” levels of peer attachment, aggressive behavior, or RESE?
The authors are also encouraged to revise their Results section in the following ways:
- If possible, provide a descriptive table that characterizes the sample beyond just age and gender.
- Explore the extent to which there were differences across junior high versus high school students on gender and any other demographic characteristics that are available. Also consider exploring whether there were between-group differences in peer attachment, RESE, and aggressive behavior.
- Define the abbreviations M1, M2, M3, …, M7 in Table 1.
- What types of correlations were conducted to produce the results in Table 1? Not all of these coefficients should represent Pearson product-moment correlations, as gender is a binary variable and would require calculation of point-biserial coefficients when considered alongside continuous independent and dependent variables.
Finally, the authors could consider make two minor changes to their Discussion:
- The authors say on page 6 that “families and schools should seek to cultivate RESE so that adolescents can actively regulate their emotions to reduce risks to healthy adolescent development.” The authors could expand upon this implication and specify/propose programs, resources, and other things that teachers and families could do to promote development of RESE skills and beliefs.
- It is customary to list limitations as the second-to-last paragraph in the Discussion, following by a summary/conclusion paragraph. Thus, the authors should slightly revise and switch up the order of their final two paragraphs, such that their second-to-last paragraph is the one that begins with “Despite these contributions, this study had some limitations” and the last one begins with “In summary, the present study makes the following contributions to the field”.
Author Response
Response to Reviewer’s Comments
The authors submitted a manuscript evaluating relationships between self-reported peer attachment, aggressive behavior, and regulatory emotional self-efficacy (RESE) within Chinese adolescents. The authors proposed two hypotheses: 1) peer attachment would negatively predict aggressive behavior, and 2) two components of RESE (managing negative emotions and expressing positive emotions) would mediate this negative relationship. To evaluate these relationships and test these hypotheses, the authors considered cross-sectional survey responses from 1108 Chinese students (549 junior high, 559 high school), calculated a series of correlation coefficients, and conducted a parallel mediation analysis. Results partially supported the authors’ hypotheses: as expected, peer attachment was negatively associated with aggressive behavior, and managing negative emotions emerged as a mediator of this relationship. However, expressing positive emotions did not mediate the relationship between peer attachment and aggressive behavior.
The authors are commended for exploring relationships between potentially impactful social behaviors during adolescence. Adolescence is considered by many to be a developmental phase in which relationships can powerfully influence emotional and behavioral outcomes, so continued multicultural research into relationships like those described in this manuscript is important and warranted. However, the manuscript as written possesses both major and minor limitations that negatively impact the paper’s suitability for publication in International Journal of Environmental Research and Public Health.
Major Limitations
First, the authors are encouraged to improve the reporting and justification behind the mediation analysis conducted in this manuscript. The following issues should be addressed:
- Conducting mediation analysis on cross-sectional data is not universally supported and could produce biased estimates of relationships and effects (see Maxwell & Cole, 2007, Psychological Methods). Because of this, drawing causal inferences from such an approach is discouraged. The authors do well to mention their cross-sectional methodology as a limitation towards the end of the Discussion and to encourage longitudinal follow-up studies. However, the manuscript also includes hints at causality in other places. One example of this occurs on page 6, where the authors state: “The results of this study also suggest that a high level of peer attachment is conducive to the development ofgood social behavior, while a low level of peer attachment can manifest as a lack of necessary trust and communication with peers, alienation, and being more prone to aggressive behaviors.” Another occurs on page 3, where the authors state “Based on the previous research, this study examines whether peer attachment can affect aggressive behaviors by first affecting RESE as a form of self-evaluation.” The authors are encouraged to critically review their manuscript and to be less forceful at inferring causal relationships from their design and findings.
Response: Thank you for the advice. Based on the previous research and theory, we conducted a cross- sectional study to investigate the role of regulatory emotional self-efficacy in the relationship between peer attachment and adolescent aggressive behavior. Firstly, according to the social network theory [1], individuals establish emotional ties with many important psychological needs, such as emotional support and affirmation. And attachment relationships indirectly affect adolescents aggressive behavior through aspects of self-evaluation (e.g., self-control, regulatory emotional self-efficacy) [2]. Thus, we believed that better RESE can more effectively respond to interpersonal conflicts, regulate negative emotions, which in turn abstain from aggressive behavior.
Secondary, as advised, we closely read the article (Maxwell & Cole, 2007), which demonstrated that mediation analysis on cross-sectional data would produce biased estimates of relationships and effects and recommended using longitudinal data for mediation analysis [3]. And researcher indicated that cross-sectional study was a common practice to explore mediating mechanism in the academic [4]. 53% of articles covering mediating mechanisms which were published in the five representative international journals (Journal of Personality and Social Psychology, Journal of Consulting and Clinical Psychology, Journal of Applied Psychology, Health Psychology and Developmental Psychology), were based on cross-sectional design [4]. Although it could not infer causal relationships from their design and findings, they also provided literature basis and research perspectives for further longitudinal studies, which were enormous in the development of psychology.
Finally, refer to your suggestion, we have reviewed our manuscript and rephrase some places as follow:
“The results of this study also suggest that a high level of peer attachment is conducive to the development of good social behavior, while a low level of peer attachment manifesting as a lack of necessary trust and communication with peers is positively associated with aggressive behavior.” “Based on previous research, this study examines whether RESE may mediate the relationship between peer attachment and aggressive behavior.”
Reference:
[1] Veenstra, R.; Dijkstra, J. K. Transformations in Adolescent Peer Networks, Laursen, B., Collins, W. A., Eds.; Sage: New York, NY, USA, 2011; Relationship pathways: From adolescence to young adulthood, pp. 135-154.
[2] Murphy, T. P.; Laible, D.; Augustine, M. The influences of parent and peer attachment on bullying. J Child Fam Stud. 2017, 26, 1388-1397, doi:10.1007/s10826-017-0663-2.
[3] Maxwell, S. E.; Cole, D. A. Bias in cross-sectional analyses of longitudinal mediation. Psychol Methods. 2007, 12(1), 23-44, doi:10.1037/1082-989X.12.1.23.
[4] Gan, Y. Q. The new trends of mediating mechanisms: research design and data statistic method. Chinese Mental Health Journal, 2014, 28(8), 584-585, doi:10.3969/j.issn.1000-6729.2014.08.005.
- The authors are strongly encouraged to produce a series of two figures that lay out the following: 1) a conceptual model of the mediation analysis that includes no coefficients/statistics, but demonstrates the expected, hypothesized direct and indirect relationships; and 2) the full mediation model that visually depicts the indirect/a and b, direct/c, and total/c’ effects described in the Results, as well as the effects of the covariates (age and gender) on the outcome.
Response: Thank you for the advice. We have added two figures in our manuscript. Figure 1 indicated that the proposed conceptual model of peer attachment, RESE and aggressive behavior (Page 3). Figure 2 indicated the mediation model, which visually depicts the direct and indirect effect of peer attachment on aggressive behaviors, as well as the effects of the covariates (age, gender, father's education level and mother's education level) on the outcome (Page 6).
Second, the authors are encouraged to more thoroughly reinforce the importance, novelty, and contribution of this manuscript to the literature and to more comprehensively answer the “So what?” question that readers may ask when reviewing their work. Some ideas on how this can be done include the following:
- The authors say towards the end of the Introduction that, although the relationship between peer attachment and aggression has been already established, “merely examining this relationship is not sufficient to create a practical application.” Application of what?
Response: Thanks for reviewer’s comment. Though some existing research have explored the relationship between peer attachment and aggressive behavior among adolescents [9, 13], the intrinsic mechanisms by which peer attachment impacted on aggressive behavior were unknow clearly. A thorough understanding of the effects of peer attachment on aggressive may conductive to develop effective interventions aimed at preventing aggressive behavior. Thus, based on previous research, this study examined whether RESE may mediate the relationship between peer attachment and aggressive behavior.
- Is there something unique about Chinese adolescents that make them particularly vulnerable to developing aggressive behavior than adolescents from other developed countries?
Response: Thanks for reviewer’s comment. In the context of Chinese culture, it emphasizes collevtive and group relationships [1]. Individuals with insecure attachments with peer have stronger emotional responses to conflicts in intimate relationships, and are more likely to be emotionally distant from others, which lead to aggressive behavior.
Reference:
[1] Chen, X. (2000). Growing up in a collectivist culture: Socialization and socioemotional development in Chinese children. In A. L. Comunian & U. P. Gielen (Eds.), International perspectives on human development (p. 331–353). Pabst Science Publishers.
- How is this manuscript advancing understanding of adolescent development within the aims and scope of this journal: environmental health sciences, public health, mental health, global health, children’s health, etc.?
Response: Thanks for reviewer’s comment. Aggressive behaviour is a major obstacle to social integration whether it be in terms of having access to certain residential settings, educational and occupational programmes or general social acceptability.
Though prior studies have indicated the direct relationship between peer attachment and adolescents’ aggressive behavior [9,13], our study has explored how peer attachment impacted adolescents’ aggressive behavior through RESE. It provides guidance to prevent adolescents from indulging or suppressing their negative emotions and from developing aggressive behaviors against others or themselves.
Minor Limitations
Several questions regarding the Materials and Methods section require clarification:
- What made a questionnaire “valid” in this particular study? In other words, what led to the reduction in sample size from 1200 to 1108?
Response: Thank you for the comment. In order to ensure the validity of the questionnaire, we will further check the respondence of participants after data collection. We are mainly due to the following reasons to exclude invalid participants: (a) some students failed to answer most of the items; (b) some students occur inconsistent responses in homogeneous items, or occur consistent responses in reverse items.
Therefore, we were distributed 1200 questionnaires and finally retrieved 1171 valid questionnaires. We have revised this description to make it more reasonable. For ease of your review, we included the added texts as follows:
“A total of 1200 adolescents participated in this study with the informed consent of school leaders, parents and the adolescents themselves. However, there were 29 invalid questionnaires due to the following reasons: (a) more than 10% missing data; and (b) some students occur inconsistent responses in homogeneous items, or occur consistent responses in reverse items. Therefore, finally retrieved 1171 valid questionnaires and included in the analyses. ”
- The authors report distributing questionnaires to “middle school students from the first to the third grade” but then report collecting data from “junior high school and high school students”. Is this a typo on the authors’ part?
Response: I’m sorry for the mistake and thanks for noticing this error. We have
revised this description in our manuscript (Page 3-4). For ease of your review, we included the added texts as follows:
“547 adolescents are junior middle school students (the number of participants of grades 7, 8, and 9 are 182, 218, and 147 respectively) and 624 adolescents are senior middle school students (the number of participants of grades 10, 11, and 12 are 205, 213, and 206 respectively). Moreover, the gender distribution of the participants in this study was 49.70% males (n = 582) and 50.30% females (n = 589). The participants ranging from 11 to 19 years old, and the average age was 14.86 years old (SD = 1.87 years old).”
- What were the months/years of data collection?
Response: Thank you for the comment. The data was collected in October 2018.
- What was the “cluster” level factor for the sampling method employed by the authors?
Response: Thank you for your question. In current study, we recruited participants from three secondary schools in southern China's Guangzhou. The sample was first stratified by the type of school: junior high schools, senior high schools and complete middle schools (complete middle schools combine junior and senior high school). And then random cluster sampling was used to randomly select two classes from each grade in each school.
- Since participants were not adults, was assent collected as a part of the protocol? And were participants compensated?
Response: Thank you for the comment. Informed consent was obtained from all subjects involved in the study. At the end of this investigation, adolescents received stationery gifts to thank them for their participation. We added some content to describe the part of Participants and procedures. For ease of your review, we included the added texts as follows:
“We recruited participants from three secondary schools in southern China's Guangzhou. The sample was first stratified by the type of school: junior high schools, senior high schools and complete middle schools (complete middle schools combine junior and senior high school). And then random cluster sampling was used to randomly select two classes from each grade in each school. The authenticity, independence and integral nature of all answers as well as the confidentiality of the information collected were emphasized to all participants by welltrained psychology graduate students. A total of 1200 adolescents participated in this study with the informed consent of school leaders, parents and the adolescents themselves. However, there were 29 invalid questionnaires due to the following reasons: (a) more than 10% missing data; and (b) some students occur inconsistent responses in homogeneous items, or occur consistent responses in reverse items. Therefore, finally retrieved 1171 valid questionnaires and included in the analyses. 547 adolescents are junior middle school students (the number of participants of grades 7, 8, and 9 are 182, 218, and 147 respectively) and 624 adolescents are senior middle school students (the number of participants of grades 10, 11, and 12 are 205, 213, and 206 respectively). Moreover, the gender distribution of the participants in this study was 49.70% males (n = 582) and 50.30% females (n = 589). The participants ranging from 11 to 19 years old, and the average age was 14.86 years old (SD = 1.87 years old).”
- How are the self-report measures scored, and what are the ranges of possible scores? Further, are there thresholds that indicate things like “high,” “low,” or “clinically significant” levels of peer attachment, aggressive behavior, or RESE?
Response: Thank you for the comment. First, in present study, participants completed anonymous questionnaires regarding the peer attachment, aggressive behavior, and RESE by self-report. All the items rate on a 5-point Likert scale. Specifically, the revised version of the Inventory of Parent and Peer Attachment rated on a 5-point Likert scale (from “1 = never” to “5 = always”); the Chinese version of the RESE scale rated on a 5-point Likert scale (from “1 = not like me at all” to “5 = very much like me”); on a 5-point Likert scale (from “1 = never” to “5 = always”); and Behavioral Tendency Questionnaire for Adolescent Behavior Problems measured on a 5-point Likert scale (from “1 = never” to “5 = always”). Therefore, the ranges of possible scores is range from 1 to 5.
Second, the mean above one standard deviation indicates high levels of peer attachment, aggressive behavior, and RESE; and the mean below one standard deviation indicates low levels of peer attachment, aggressive behavior, and RESE.
The authors are also encouraged to revise their Results section in the following ways:
- If possible, provide a descriptive table that characterizes the sample beyond just age and gender.
Response: Thank you for your suggestion. We added some content to describe the sample characteristic (Page 3-4). For ease of your review, we included the added texts as follows:
“ 547 adolescents are junior middle school students (the number of participants of grades 7, 8, and 9 are 182, 218, and 147 respectively) and 624 adolescents are senior middle school students (the number of participants of grades 10, 11, and 12 are 205, 213, and 206 respectively). Moreover, the gender distribution of the participants in this study was 49.70% males (n = 582) and 50.30% females (n = 589). The participants ranging from 11 to 19 years old, and the average age was 14.86 years old (SD = 1.87 years old).”
- Explore the extent to which there were differences across junior high versus high school students on gender and any other demographic characteristics that are available. Also consider exploring whether there were between-group differences in peer attachment, RESE, and aggressive behavior.
Response: Thank you for your suggestion.
Table 2 shows group differences of the variables in this study between junior school versus high school students. Independent-samples t tests found that high school students’s age was significantly higher than junior school students’ age. And Father's and mother’s education level among junior school students were significantly higher than that of high school students. However, there were no significant difference in peer attachment, POS, NEG, and aggressive behavior between junior school versus high school students.
Table 2 Group differences of the variables in this study between junior school versus high school students
|
Variables |
Junior school students |
High school students |
t |
||
|
Mean |
SD |
Mean |
SD |
||
|
Age |
13.18 |
1.00 |
16.34 |
1.02 |
-53.25*** |
|
Father's education level |
2.62 |
0.74 |
2.37 |
0.72 |
5.84*** |
|
Mother's education level |
2.46 |
0.78 |
2.15 |
0.71 |
7.08*** |
|
Peer attachment |
2.78 |
0.64 |
2.78 |
0.57 |
0.08 |
|
POS |
4.01 |
0.83 |
4.09 |
0.73 |
-1.67 |
|
NEG |
3.42 |
0.76 |
3.39 |
0.71 |
0.77 |
|
Aggressive behavior |
1.54 |
0.81 |
1.62 |
0.83 |
-1.55 |
Note: N=1171. POS=self-efficacy in expressing positive emotions, NEG=self-efficacy in managing negative emotions. SD = standard deviation. *p < .05; **p < .01; ***p < .001.
- Define the abbreviations M1, M2, M3, …, M7 in Table 1.
Response: Thank you for your suggestion.
The results of the mediation analysis presented in Table 2. As is shown in Model 2, peer attachment could significantly predict POS (b = .49, p < .001). And as shown in Model 4, peer attachment could significantly predict NEG (b = .40, p < .001). And as shown in Model 6, peer attachment significantly predict aggressive behavior (b = -.20, p < .001). Moreover, as shown in Model 7, after controlling POS and NEG, peer attachment was less predictive of aggressive behavior (b = -.13, p < .05). And only NEG could significantly predict aggressive behavior (b = -.20, p < .001). The mediation model presented in Figure 2.
- What types of correlations were conducted to produce the results in Table 1? Not all of these coefficients should represent Pearson product-moment correlations, as gender is a binary variable and would require calculation of point-biserial coefficients when considered alongside continuous independent and dependent variables.
Response: Thanks for reviewer’s comment. We virtualized gender, which was dummy-coded as 0 (= female) and 1 (= male). Therefore, all of these coefficients can conducted Pearson product-moment correlations. Table 1 is the result of Pearson product-moment correlations.
Finally, the authors could consider make two minor changes to their Discussion:
- The authors say on page 6 that “families and schools should seek to cultivate RESE so that adolescents can actively regulate their emotions to reduce risks to healthy adolescent development.” The authors could expand upon this implication and specify/propose programs, resources, and other things that teachers and families could do to promote development of RESE skills and beliefs.
Response: Thanks for reviewer’s comment. We have expanded the implication in our article. The current study found that RESE mediated the relationship between peer attachment and aggressive behavior. It may be difficult to improve or influence individuals' attachment over time, but there are effective measures to enhance individuals' RESE, so that they can better regulate and manage their emotions. For educational practice, courses or activities on emotion education should be used as a vehicle to convey emotion regulation knowledge and skills to adolescents, guide them to learn to regulate their own emotions, and gradually enhance their RESE. When students, under the guidance of teachers and parents, are able to face and accept their emotions, try to describe their inner feelings in words, learn to broaden their horizons and reinterpret the meaning of certain events, it enables adolescents to take control of their emotions. They then become more confident in their ability to regulate their emotions. The increase in adolescents' RESE not only facilitates the improvement of their emotional state, but more importantly, through the gradual increase in RESE, college students learn to gain a sense of control over their daily emotional experiences and increase their self-confidence, which in turn extends this good feeling to other aspects of self-understanding and decrease externalizing behaviors.
- It is customary to list limitations as the second-to-last paragraph in the Discussion, following by a summary/conclusion paragraph. Thus, the authors should slightly revise and switch up the order of their final two paragraphs, such that their second-to-last paragraph is the one that begins with “Despite these contributions, this study had some limitations” and the last one begins with “In summary, the present study makes the following contributions to the field”.
Response: Thank for the advice. According to your suggestion, we switched up the order of final two paragraphs and add a new paragraph of conclusion as follow:
“5. Implication
In summary, the present study makes the following contributions to the field. First, it constructs and confirms a mediating model with RESE as an intermediary, which reveals the development mechanism of adolescents’ aggressive behavior. Second, in contrast to most existing research on mental health and peer influence, which starts from the perspective of poor peer behavior learning, this study focuses on the emotions of adolescents and explores aggressive behavior in adolescents using the concept of peer attachment. Third, it demonstrates that RESE plays an indispensable role in adolescent development. This is important since past research has emphasized emotional regulation but it neglected adolescents’ self-evaluation of their self-regulation ability. This strongly suggests that we must actively cultivate adolescents’ RESE.
- Iimitation
Despite these contributions, this study has several limitations. First, although the reliability and validity analysis report showed that the scales met the requirements of psychological statistics, it is difficult to avoid the reliability problem caused by self-reporting. Future research should try to collect data from multiple informants and methods (e.g. teacher assessment and peer nomination). Second, the cross-sectional design cannot fully reveal the causal relationship between peer attachment and externalization of problem behaviors by adolescents. Therefore, future research should use vertical tracking technology to explore the relationship between peer attachment and aggressive behavior. Third, the present study focus on the unidirectional relationship between peer attachment and aggressive behaviors. However, Charalampous et al. [41] found that peer attachment would influence bullying/victimization among adolescent. Therefore, it is meaningful to examine the bidirectional effect of peer attachment and aggressive behaviors in future research. Finally, this study considered only one mediator (i.e. RESE), future research should explore other mediators or moderator on the association between peer attachment and aggressive behaviors.
- Conclusion
Taken together, the current study demonstrated that peer attachment was a crucial factor to adolescent aggressive behavior via regulatory emotional self-efficacy. The finding highlighted that educators were ought to consider the importance role of peer attachment and regulatory emotional self-efficacy during designing prevention of aggressive behavior.”

Reviewer 2 Report
This is an interesting study on the peer attachment, regulatory emotional self-efficacy, and aggressive behaviors among Chinese adolescents.
Before the publication the manuscript needs a few improvements:
- In some places the text becomes hardly comprehensible because of linguistic and grammatical mistakes, example:
"This scale has been previously shown adequate psychometric properties when applied to Chinese people"
I would recommend to check the text with a support of a native speaker or an English language specialist. - I would suggest to add some newer citations to the Introduction to make the references more up to date. Examples:
Vagos P, Carvalhais L. The Impact of Adolescents' Attachment to Peers and Parents on Aggressive and Prosocial Behavior: A Short-Term Longitudinal Study. Front Psychol. 2020 Dec 23;11:592144.
Malonda E, Llorca A, Mesurado B, Samper P, Mestre MV. Parents or Peers? Predictors of Prosocial Behavior and Aggression: A Longitudinal Study. Front Psychol. 2019 Oct 22;10:2379.
Author Response
Response to Reviewer’s Comments
This is an interesting study on the peer attachment, regulatory emotional self-efficacy, and aggressive behaviors among Chinese adolescents.
Before the publication the manuscript needs a few improvements:
Point 1: In some places the text becomes hardly comprehensible because of linguistic and grammatical mistakes, example:
"This scale has been previously shown adequate psychometric properties when applied to Chinese people"
I would recommend to check the text with a support of a native speaker or an English language specialist.
Response 1: Thanks for reviewer’s comment. We have checked the text with a support of an English language specialist, example:
“This scale has been previously shown to have good reliability and validity when used in Chinese adolescents [17].”
Point 2: I would suggest to add some newer citations to the Introduction to make the references more up to date. Examples:
Vagos P, Carvalhais L. The Impact of Adolescents' Attachment to Peers and Parents on Aggressive and Prosocial Behavior: A Short-Term Longitudinal Study. Front Psychol. 2020 Dec 23;11:592144.
Malonda E, Llorca A, Mesurado B, Samper P, Mestre MV. Parents or Peers? Predictors of Prosocial Behavior and Aggression: A Longitudinal Study. Front Psychol. 2019 Oct 22;10:2379.
Response 2: Thank you for the advice very much. We have revised and supplemented the references in the manuscript.

Reviewer 3 Report
Methods: The authors should specify more in detail the characteristic and the choice of the sample. In the description of the sample we read that they did a selection but it'sn not clear what are the criteria used. It's also necessary to specify if the authors have the consent to participate to the research by the subjects and by their parents. It's not clarify if the questionnaire was completed in written form and who did the submission: experimenter or other people and it's not celar if thery are or not present during the compilation of it. At last but not least do the experimeters are trained people?
The bibliography report researchs till 2016 but the study concerns very actual topics so the authors should have more recent citations. This could improve theoretical background too and it could add other research hypothesis to the already present
Author Response
Response to Reviewer’s Comments
Point 1: Methods: The authors should specify more in detail the characteristic and the choice of the sample. In the description of the sample we read that they did a selection but it'sn not clear what are the criteria used. It's also necessary to specify if the authors have the consent to participate to the research by the subjects and by their parents. It's not clarify if the questionnaire was completed in written form and who did the submission: experimenter or other people and it's not celar if thery are or not present during the compilation of it. At last but not least do the experimeters are trained people?
Response 1: Thanks for reviewer’s comment. We have rewritten the methods in our article as bellow.
“We recruited participants from three secondary schools in Guangzhou, China. Random cluster sampling was used to randomly select two classes in each grade of the schools. The authenticity, independence and integral nature of all answers as well as the confidentiality of the information collected were emphasized to all participants by welltrained psychology graduate students. A total of 1200 adolescents participated in this study with the informed consent of school leaders, parents and the adolescents themselves. However, there were 29 invalid questionnaires due to the following reasons: (a) more than 10% missing data; and (b) some students occur inconsistent responses in homogeneous items, or occur consistent responses in reverse items. Therefore, finally retrieved 1171 valid questionnaires and included in the analyses. 547 adolescents are junior middle school students (the number of participants of grades 7, 8, and 9 are 182, 218, and 147 respectively) and 624 adolescents are senior middle school students (the number of participants of grades 10, 11, and 12 are 205, 213, and 206 respectively). Moreover, the gender distribution of the participants in this study was 49.70% males (n = 582) and 50.30% females (n = 589). The participants ranging from 11 to 19 years old, and the average age was 14.86 years old (SD = 1.87 years old).
Point 2: The bibliography report researchs till 2016 but the study concerns very actual topics so the authors should have more recent citations. This could improve theoretical background too and it could add other research hypothesis to the already present.
Response 2: Thanks for reviewer’s comment. We have added more recent citations, as follow:
- Malonda, E.; Llorca, A.; Mesurado, B.; Samper, P.; Mestre, M. V. Parents or peers? Predictors of prosocial behavior and ag-gression: A longitudinal study. Front.Psychol. 2019, 10, 2379, doi:10.3389/fpsyg.2019.02379.
- Saladino, V.; Mosca, O.; Lauriola, M.; Hoelzlhammer, L.; Cabras, C.; Verrastro, V. Is family structure associated with deviance propensity during adolescence? The role of family climate and anger dysregulation. Int. J. Environ.Res. Public Health. 2020, 17, 9257, doi:10.3390/ijerph17249257.
- Tian, Y.; Yu, C.; Lin, S.; Lu, J.; Liu, Y.; Zhang, W. Parental psychological control and adolescent aggressive behavior: Deviant peer affiliation as a mediator and school connectedness as a moderator. Front.Psychol. 2019, 10, 358, doi:10.3389/fpsyg.2019.00358.
- Lin, S.; Yu, C.; Chen, J.; Zhang, W.; Cao, L.; Liu, L. Predicting adolescent aggressive behavior from community violence exposure, deviant peer affiliation and school engagement: A one-year longitudinal study. Child Youth Serv Rev. 2020, 111, 104840, , do:10.1016/j.childyouth.2020.104840.
- Vagos, P.; Carvalhais, L. The impact of adolescents' attachment to peers and parents on aggressive and prosocial behavior: A short-ierm longitudinal study. Front Psychol. 2020, 11, 592144, , doi:10.3389/fpsyg.2020.592144.
- Murphy, T. P.; Laible, D.; Augustine, M. The influences of parent and peer attachment on bullying. J Child Fam Stud. 2017, 26, 1388-1397, doi:10.1007/s10826-017-0663-2.
- Li, C.; Wang, Y.; Liu, M.; Sun, C.; Yang, Y. Shyness and subjective well-being in Chinese adolescents: Self-efficacy beliefs as mediators. J Child Fam Stud. 2020, 29, 3470-3480, doi:10.1007/s10826-020-01823-0.
- Bronfenbrenner, U. Toward an experimental ecology of human development. Am Psychol. 1979, 32, 513–531, doi:10.1037/0003-066X.32.7.513.

Reviewer 4 Report
Summary: The Authors test the relationship between peer attachment and aggressive behaviors in a large sample of Chinese adolescents using surveys. They test whether there is a direct relationship between these two variables, as well as whether regulatory emotional self-efficacy mediates the relationship. They find evidence for a direct inverse relationship between peer attachment and aggression and that the ability to regulate negative emotions, but not positive emotions, mediates this relationship. I find these results very interesting and I agree that this finding emphasizes the
importance of fostering good peer relationships. I found the introduction easy to follow and it seems well grounded in the literature (though I'm not an expert in this subfield so I can't make a strong assessment of whether there are missing elements). I have a few specific questions and concerns regarding transparency of methods and analysis and the appropriateness of the conclusions that I elaborate on below.
Specific comments:
1. I would like to see more detailed descriptions of the self-report measures used in the analyses. Example statements and whether there are subscales in the questionnaire measures would be very helpful. In particular I'd like to know more about the aggression measure. I was unable to find the measure via google scholar and the DOI/link in the reference section appears to be broken. I would imagine that, depending on the type of aggression being measured, the hypotheses about what role peer attachment and emotional self-regulatory ability would be somewhat different. For example,
if it is measuring relational aggression or instrumental aggression with the goal of social dominance or to improve standing in one's ingroup, then you might make the opposite prediction between
peer attachment and aggression. I don't think that this is likely the type of aggression being measured here, but my point is that, without a more granular description of what "aggression" refers to here,
it's hard to know exactly what is being predicted and why regulatory emotional self-efficacy would mediate this role.
2. In the regression/mediation analyses, age and gender are entered as "control" variables but I would like to see A) a justification for doing so, and B) the same analyses with these covariates removed.
Though I think there's good reason to think that gender or age may influence and even interact with some of the other model terms in predicting aggression, it's not mentioned at all in the manuscript. As pointed out in this review by Simmons, Nelson, & Simonsohn, 2011 (https://journals.sagepub.com/doi/full/10.1177/0956797611417632)
the addition of covariates to a model can increase the likelihood of a false positive result. So, consistent with the recommendations of that paper and for the sake of analytical transparency, I'd like to see whether these analyses hold without the inclusion of age and gender as covariates. If the results diverge without these covariates, I think the Authors should do more to justify their inclusion and comment on potential limitations of the results. If they remain the same, it is good to know that the relationship
is not contingent upon controlling for these two factors.
3. Related to the above, I'm having a hard time visualizing how the mediation analyses are conducted while including these covariates. I believe the Authors are using the Baron & Kenny mediation approach (though this clarification would also be helpful!) and typically these mediation relationships are visually depicted as in diagrams with an X-->Y, X-->M, and M-->Y. I'm unclear where the covariates come into play in this approach, so a visualization here would be very helpful.
4. Though it is mentioned at the very end of the discussion, I think the limitation about determining causality from this correlational relationship needs to be brought up sooner and addressed in much more detail, ideally supplemented with some additional analyses.
I would think that it's just as plausible for the effect to work in the opposite direction. That is, being less aggressive leads to stronger peer relationships. It may be useful to show whether the same analysis with the directionality reversed (X = aggression, Y = peer attachment, and M = RESE) and other combinations where peer attachment or aggression are the mediators show divergent results. There are also different forms of mediation (such as causal mediation analysis) that would be better suited to this question that I think would greatly strengthen the paper and claims if conducted.
Author Response
Response to Reviewer’s Comments
Summary: The Authors test the relationship between peer attachment and aggressive behaviors in a large sample of Chinese adolescents using surveys. They test whether there is a direct relationship between these two variables, as well as whether regulatory emotional self-efficacy mediates the relationship. They find evidence for a direct inverse relationship between peer attachment and aggression and that the ability to regulate negative emotions, but not positive emotions, mediates this relationship. I find these results very interesting and I agree that this finding emphasizes the importance of fostering good peer relationships. I found the introduction easy to follow and it seems well grounded in the literature (though I'm not an expert in this subfield so I can't make a strong assessment of whether there are missing elements). I have a few specific questions and concerns regarding transparency of methods and analysis and the appropriateness of the conclusions that I elaborate on below.
Specific comments:
Point 1: I would like to see more detailed descriptions of the self-report measures used in the analyses. Example statements and whether there are subscales in the questionnaire measures would be very helpful. In particular I'd like to know more about the aggression measure. I was unable to find the measure via google scholar and the DOI/link in the reference section appears to be broken. I would imagine that, depending on the type of aggression being measured, the hypotheses about what role peer attachment and emotional self-regulatory ability would be somewhat different. For example, if it is measuring relational aggression or instrumental aggression with the goal of social dominance or to improve standing in one's ingroup, then you might make the opposite prediction between peer attachment and aggression. I don't think that this is likely the type of aggression being measured here, but my point is that, without a more granular description of what "aggression" refers to here, it's hard to know exactly what is being predicted and why regulatory emotional self-efficacy would mediate this role.
Response 1: Thank you for the advice very much. We adopt the questionnaire was developed by Zhang et al. [33] and is a Chinese measuring tools. The 6-item questionnaires was used to evaluate aggressive behavior (include self-harm and article damage) and no subscales. For all items, adolescents need to report: (1) When I am in a bad mood, I get rid of it by hurting myself; (2) I often deliberately hurt myself; (3) When I get upset, I do something out of line; (4) I lost my temper and broke things for no reason; (5) I often hit the wall or something like that when I’m bored; 6) I don’t think it’s wrong to drop things when you’re in a bad mood.
We have added more detailed descriptions in the part of Measure. For ease of your review, we included the added texts as follows:
“The aggression tendency dimension of the Behavioral Tendency Questionnaire for Adolescent Behavior Problems compiled by Zhang et al. [33] was used to measure adolescents’ aggressive behavior during past 6 months. A sample item is “I often hit the wall or something like that when I'm bored”. This questionnaire consists of 6 items measured on a 5-point Likert scale (from “1 = never” to “5 = always”), where a high score indicates a strong aggression tendency. The Cronbach’s alpha for the aggression tendency dimension is .83.”
Point 2: In the regression/mediation analyses, age and gender are entered as "control" variables but I would like to see A) a justification for doing so, and B) the same analyses with these covariates removed.
Though I think there's good reason to think that gender or age may influence and even interact with some of the other model terms in predicting aggression, it's not mentioned at all in the manuscript. As pointed out in this review by Simmons, Nelson, & Simonsohn, 2011 (https://journals.sagepub.com/doi/full/10.1177/0956797611417632)
the addition of covariates to a model can increase the likelihood of a false positive result. So, consistent with the recommendations of that paper and for the sake of analytical transparency, I'd like to see whether these analyses hold without the inclusion of age and gender as covariates. If the results diverge without these covariates, I think the Authors should do more to justify their inclusion and comment on potential limitations of the results. If they remain the same, it is good to know that the relationship is not contingent upon controlling for these two factors.
Response 2: Gender and age are entered as control variables in all regression models.
Point 3: Related to the above, I'm having a hard time visualizing how the mediation analyses are conducted while including these covariates. I believe the Authors are using the Baron & Kenny mediation approach (though this clarification would also be helpful!) and typically these mediation relationships are visually depicted as in diagrams with an X-->Y, X-->M, and M-->Y. I'm unclear where the covariates come into play in this approach, so a visualization here would be very helpful.
Response 3: Gender and age are entered as control variables in all regression models.
The reviewer understand it wrong. In this study, we haven’t use Baron & Kenny mediation approach. Bias-corrected bootstrap confidence intervals were used to test for significance of the direct and indirect paths.
Point 4: Though it is mentioned at the very end of the discussion, I think the limitation about determining causality from this correlational relationship needs to be brought up sooner and addressed in much more detail, ideally supplemented with some additional analyses.
I would think that it's just as plausible for the effect to work in the opposite direction. That is, being less aggressive leads to stronger peer relationships. It may be useful to show whether the same analysis with the directionality reversed (X = aggression, Y = peer attachment, and M = RESE) and other combinations where peer attachment or aggression are the mediators show divergent results. There are also different forms of mediation (such as causal mediation analysis) that would be better suited to this question that I think would greatly strengthen the paper and claims if conducted.
Response 4: Thank you for the advice very much. It is interesting and meaningful to explore reversed relationship and other combinations of these variable. However, in order to clear the risk or protective factors of aggressive behavior among adolescents, we argue that should focus on the impact factor and underlying mechanism. Therefore, the current study aim to explore the impact factor and the mediating mechanisms of aggressive behaviors among Chinese adolescents. Based on theory and previous studies, we regard peer attachment as independent variable, RESE as mediating variable, and focus on the direct and indirect association between peer attachment and aggressive behaviors.
We have corrected some inaccurate terms when describe causality in the manuscript. In addition, we have supplemented some content to further explain the limitation of our study (Page 8). For ease of your review, we included the added texts as follows:
“Third, the present study focus on the unidirectional relationship between peer attachment and aggressive behaviors. However, Charalampous et al. [41] found that peer attachment would influence bullying/victimization among adolescent. Therefore, it is meaningful to examine the bidirectional effect of peer attachment and aggressive behaviors in future research. Finally, this study considered only one mediator (i.e. RESE), future research should explore other mediators or moderator on the association between peer attachment and aggressive behaviors.”

Round 2
Reviewer 4 Report
The Authors have responded appropriately to points 1 and 4 and I am satisfied with their answers. However, their responses to points 2 and 3 do not address my concerns. I would like to see these points actually addressed in more detail than simply re-stating what is already mentioned in this paper.
For point 2, I believe it is important to know whether the effects hold when not controlling for age and gender. For the sake of scientific transparency and for the interpretation of the results to be accurate, it would be good to know whether or not the results are contingent upon controlling for these factors or not. I request that the Authors please report the analyses both ways and if there are discrepancies, address them in the manuscript.
For point 3, if I misunderstood how the mediations were conducted, then I request that the Authors please provide more detailed information. Again, I think a graphical depiction would be useful here to understand what the indirect and direct paths are and how the control variables fit into it.
Author Response
Thank very much for your constructive suggestions. We modified and checked our manuscript in detail.
For point 2, we conducted two mediating effect analysis when controlling for covariates and when not controlling for covariates. The results showed that the mediating effects of NEG between peer attachment and adolescents’ aggressive behavior were significant whether control covariates or not. In particular, the mediating effect size is -.08, 95% CI [-.11, -.05] when controlling for covariates. Moreover, the mediating effect size is -.06, 95% CI [-.10, -.03] when not controlling for covariates. The results also showed that the mediating effect of POS between peer attachment and adolescents’ aggressive behavior were significant when not controlling for covariates (b = -0.03, 95% CI [-.08, -.002]); However, this mediating effect was nonsignificant when controlling for covariates (b = -0.02, 95% CI [-.05, .02]).
For point 3, we conducted mediation analysis by using Model 4 of Hayes’s PROCESS macro [34] to examine the mediating effect of self-efficacy in expressing positive emotions and self-efficacy in managing negative emotions. Following the methods of previous studies [24], we used Four-step procedure to establish mediation effect, which requires (a) a significant association between peer attachment and self-efficacy in expressing positive emotions (POS); (b) significant association between peer attachment and self-efficacy in managing negative emotions (NEG); (c) significant association between POS, NEG and aggressive behavior while controlling for peer attachment; and (d) significant coefficient for the indirect path between peer attachment and aggressive behavior via POS and NEG. Compute the 95% confidence interval (CI) by using 5000 bootstrap samples to determines whether the last condition is satisfied. A CI that did not contain 0 indicated a significant mediating effect on the mediation path. In all analyses, we included adolescents’ age, gender, father’s education level, and mother's education level as covariates.